# Histochemical and Immunohistochemical Evaluation of the Effects of a Low-Input Diet on Different Chicken Breeds

**DOI:** 10.3390/ani15050696

**Published:** 2025-02-27

**Authors:** Elisa Fonsatti, Martina Bortoletti, Marco Birolo, Francesco Bordignon, Gerolamo Xiccato, Angela Trocino, Daniela Bertotto, Marta Vascellari, Giuseppe Radaelli, Cristina Ballarin

**Affiliations:** 1Department of Comparative Biomedicine and Food Science (BCA), University of Padova, Viale dell’Università, 16, 35020 Legnaro, PD, Italy; martina.bortoletti@unipd.it (M.B.); daniela.bertotto@unipd.it (D.B.); cristina.ballarin@unipd.it (C.B.); 2Department of Agronomy, Food, Natural Resources, Animals and Environment (DAFNAE), University of Padova, Viale dell’Università, 16, 35020 Legnaro, PD, Italy; marco.birolo@unipd.it (M.B.); francesco.bordignon@unipd.it (F.B.); gerolamo.xiccato@unipd.it (G.X.); angela.trocino@unipd.it (A.T.); 3Histopathology Laboratory, Istituto Zooprofilattico Sperimentale delle Venezie, 35020 Legnaro, PD, Italy; mvascellari@izsvenezie.it

**Keywords:** low-input diets, local chicken breeds, gut morphology, immunohistochemistry evaluation

## Abstract

The environmental impact of poultry farming can be reduced by using diets made from locally sourced raw materials and by raising local chicken breeds. These breeds are better adapted to environmental changes and need less intensive management, making them ideal for sustainable farming practises. In this study, we compared how different chicken breeds and their crossbreeds responded to two types of diets: a standard diet and a low-input diet that replaced part of the soybean content with local ingredients (including fava beans and GMO-free soybeans). The results showed that local breeds, particularly *Robusta maculata*, had better gut health, with structures that support efficient nutrient absorption compared to fast-growing commercial breeds like Ross. While fast-growing breeds had more mucus-producing cells, indicating a higher sensitivity to environmental and dietary stress, local breeds, like *Robusta maculata*, exhibited more resilience. The low-input diet slightly reduced gut efficiency but tended to enhance immune cell activity. These findings suggest that using local breeds and low-input diets can help maintain animal welfare and productivity while making poultry farming more sustainable.

## 1. Introduction

The environmental impact of human activities is of significant concern and the European Union has prioritized its reduction by drawing up a list of initiatives and measures within the European Green Deal [1]. Achieving a climate-neutral Europe includes multiple goals, among them a transition to sustainable food production [1,2]. An initiative that concerns animal husbandry and production is the “From Farm to Fork Strategy”, which outlines plans to establish fair, healthy, and environmentally sustainable food systems. Additionally, preserving and, where possible, restoring biodiversity is a central objective [2]. Agricultural activities, including meat production, are essential to meet the needs of the world’s growing population [3,4] and therefore, must be carefully evaluated for their impact [2]. Poultry production plays a key role in meeting the demand for food; the request for poultry meat and eggs is expected to double in the coming decades due to their advantageous nutritional properties, efficiency, and costs [5]. Poultry meat and eggs provide a high-quality and affordable protein source among livestock meats [6]. In terms of sustainability, poultry farming is widely recognized as the most efficient and sustainable compared to other animal production systems [7]. Although poultry farming has a relatively low environmental impact, its sustainability can and should be improved, especially given the high demand for these products [8]. The poultry industry uses fast-growing broiler chicken genotypes that perform well in terms of feed intake, feed efficiency, and growth rate. However, some welfare issues have emerged, including increased incidence of diseases [9] and reduced adaptability to environmental challenges, such as heat stress, and alternative farming systems [10]. While local and more rustic breeds exhibit lower growth rates, they adapt better to environmental changes and require less intensive management than fast-growing chickens [11]. Additionally, more rustic chicken breeds or different crossbreeding strategies allow for low-input, locally sourced diets that may not be suitable for fast-growing commercial poultry lines [10,12]. Low-input diets are typically formulated with local vegetables and legumes, which help to reduce environmental impact and do not possess the high energy content of industrial formulations [13]. In line with the goals of the European Green Deal, reducing the environmental impact of poultry farming through low-input diets and alternative chicken breeds is a promising approach. Local breeds tend to be more resilient and contribute to biodiversity [14], and crossbreeding could offer a solution to chicken production [15,16]. The local breeds considered in this study were *Bionda piemontese* (BP) and *Robusta maculata* (RM). *Bionda piemontese*, originating from the Piemonte region in Italy, is a slow-growing, late-maturing breed [17]. Although primarily farmed for meat production, it is considered a dual-purpose breed [18]. *Robusta maculata* is a cross between Tawny Orpingtons and White Americans and is used for both meat and egg production [19]. Hendrix Genetics produces a meat-type chicken known as the Sasso in France, which has been brought to market. The Sasso chickens are renowned for their medium growth rate, medium size, and soft flesh. Additionally, due to its recessive plumage traits, this chicken genotype is frequently utilized in crossbreeding programmes with regional breeds [20]. Due to its exceptional development rate and feed efficiency, the Ross 308 is the most popular genotype of fast-growing chickens for the purpose of meat production worldwide [21]. Some studies show that, in terms of performance and carcass weight, local chicken breeds are less sensitive to alternative diets [16]. The aim of this study is to analyze the responses of a commercial fast-growing genotype, local breeds, and the crosses of local breeds with a medium-growth commercial strain to a low-impact diet, with a particular focus on the effects on gut morphology and gut inflammatory responses across the different genotypes. The results of this trial could provide further valuable insights into which genotypes are more resilient.

This study is part of a larger trial, and the results of growth curve dynamics have already been previously published [22].

## 2. Materials and Methods

### 2.1. Animals, Facilities, and Experimental Design

This study was approved by the Ethical Committee for Animal Experimentation (Organismo Preposto al Benessere degli Animali, OPBA) of the University of Padova, Italy (Prot. N. 15481, approved on 1 February 2021). All animals were handled according to the principles stated in the EC Directive 2010/63/EU regarding the protection of animals used for experimental and other scientific purposes. The research staff involved in animal handling were animal specialists (PhD or MS in Animal Science) and veterinary practitioners.

The trial was conducted in the experimental poultry house facilities of the University of Padova (Italy) where 40 pens were used in which a total of 441 chickens were allocated according to a three-factorial experimental arrangement with five genotypes and two sexes (Ross, 51 females and 51 males; *Bionda piemontese* BP, 37 and 39; *Robusta maculata* RM, 25 and 47; BP × Sasso, 49 and 48; and RM × Sasso, 47 and 47) and two diets (standard diet: metabolizable energy, ME 3348 kcal/kg; crude protein, CP 18.5%; low-input diet: ME 3084 kcal/kg; CP 16.7%), i.e., two replicate pens per experimental group (genotype/sex/diet) [22]. The animals were fed a starter diet until 20 days of age; the details of the starter diet were described by Menchetti et al. [22]. The experimental diets were fed from 20 days of age until slaughtering (47 days for Ross and 105 days for other genotypes). In the low-input diet, imported genetically modified soybean (GMO-soybean) meal was partly replaced by local ingredients, i.e., faba bean (*Vicia faba*, var minor) and GMO-free organic soybean meal. Further details about diets and the other recordings were presented by Menchetti et al. [22]. The different slaughter ages for Ross (47 days) and the other genotypes (105 days) were determined based on their distinct growth rates and physiological development [22]. Given the experimental conditions in this study, extending the trial duration for Ross beyond its typical commercial slaughter age was too challenging. Therefore, Ross 308 was slaughtered at 47 days, aligning with standard industry practises and achieving approximately 55% of its adult body weight. Conversely, local breeds and their crossbreeds exhibit significantly slower growth rates, requiring more time to reach physiological maturity. These birds were slaughtered at 105 days, ensuring they reached a comparable maturity level of 55–65% of their adult body weight [22].

### 2.2. Sampling of Jejunum Tissues and Histological and Immunohistochemistry Analyses

Two days before commercial slaughtering, 6 males per genotype per experimental diet were used to sample jejunum mucosa and perform the analyses required for this study. A sample (about 2 cm in length) was dissected from the jejunum, halfway between the end of the duodenal loop and Merckel’s diverticulum [23]. The samples were washed with phosphate-buffered saline (PBS). The samples (about 1 cm) were fixed in paraformaldehyde (diluted in 0.1 M PBS, pH 7.4) for 48 h, dehydrated, and embedded in paraffin, before being submitted for immunohistochemical and histological examinations.

From each sample, 4 serial sections of 4 μm were cut and placed on four slides each one used for different staining, i.e., hematoxylin/eosin; Alcian blue (pH 2.5)-PAS; for immunohistochemical analysis (two slides). The images were acquired with Aperio LV1 (Leica Biosystems Italia, Buccinasco, Italy). The sections stained with hematoxylin/eosin were used for morphometric evaluation, measuring the length of villi and the depth of the associated crypts, using image analysis software Aperio ImageScope (Vers. 12.3.3) (Leica Biosystems Italia, Buccinasco, Italy). For each animal, 20 villi and 20 crypts were measured as described by Hampson (1986) [24]. The goblet cells that were positive for Alcian blue-PAS were counted on 10 villi per animal along 300 μm of the villus length. Automated immunohistochemistry was performed on the Discovery ULTRA system (Ventana Medical Systems Inc., Tucson, AZ, USA). Briefly, sections mounted onto superfrost plus slides were deparaffinized in an aqueous-based detergent solution (Discovery Wash, Ventana Medical Systems Inc., Tucson, AZ, USA), and subjected to heat-induced antigen retrieval (pH 8.4) for 40 min. The CD45 primary antibody (polyclonal, SouthernBiotech, Homewood, AL, USA, code 8270-01) at 1:40 dilution and the CD3 antibody (monoclonal, Agilent Dako, Santa Clara, CA, USA, code A0452) at 1:100 dilution, were applied to detect CD45+ intraepithelial leukocytes, including T-cells and B-cells, and CD3+ intraepithelial T-cells in the jejunal mucosa. After detection, sections were counterstained with Mayer’s hematoxylin (Hematoxylin II, Ventana Medical Systems Inc., Tucson, AZ, USA) and mounted with Eukitt (Kaltek, Padua, Italy). Using a reference rectangle with the short side at 100 μm, intraepithelial leukocytes in the villi were counted and represented as the density of CD45+ and CD3+ cells (expressed as cells/10,000 μm^2^). The count was performed with 10 different areas per animal. The analysis was performed by two independent observers using ImageJ (Vers. 1.53t) [25].

### 2.3. Statistical Analysis

Data from morphological analysis and density of CD45+ and CD3+ cells were submitted to ANOVA with genotype, diet, and interactions as the main effects using the PROC GLM procedure of SAS (Vers. 9.4) (Statistical Analysis System, SAS Institute Inc., Cary, NC, USA). Mean differences that were *p* ≤ 0.05 were considered statistically significant.

## 3. Results

### Gut Morphology and Immuno-Histochemical Analyses

In the morphology analysis, significant differences were observed among the genotypes at jejunum (Table 1). The longest villi were recorded in RM chickens (1316 µm), significantly exceeding those of Ross (1028 µm), BP (963 µm), and BP × SA (1016 µm) (*p* < 0.001). The villi length in RM × SA chickens (1167 µm) was comparable to RM chickens of the other genotypes. Crypts were significantly deeper in Ross chickens (136.4 µm) compared to the other genotypes, which ranged from 88.2 to 106.4 µm (*p* < 0.001). Regarding the villus-to-crypt ratio, RM chickens displayed the highest ratio (14.75) compared with RM × SA (11.19), BP (11.08), and BP × SA (10.47), which were similar among them, while Ross chickens had the lowest ratio (7.80) (*p* < 0.001). Additionally, goblet cell density (shown in Figure 1A,B) varied significantly among the genotypes (*p* < 0.001) (Table 1) where Ross chickens exhibited the highest density (21.6) compared to all the other genotypes. The densities of goblet cells in BP (19.2), BP × SA (19.4), RM (17.7), and RM × SA (17.9) were comparable to each other.

Dietary treatment also significantly influenced the intestinal morphology (Table 1). Chickens fed the low-input diet exhibited a reduction in villi height (1147 µm vs. 1049 µm, *p* = 0.05) and in villi height/crypt depth ratio (11.7 vs. 10.4, *p* < 0.05) compared to those receiving the standard diet. Additionally, the former chickens also showed a tendency to increase the density of CD3+ cells compared to chickens fed the standard diet (3092 vs. 3447 cells/µm^2^, *p* < 0.10) (Figure 1C,D). No variations were observed in CD45+ cell density between genotypes or diets (Figure 1E,F). The interaction between genotypes and diet (Table 1) showed no significant differences.

## 4. Discussion

This study provides results concerning the impact of genotype and diet on chicken gut morphology and intestinal immune response. Taller villi, as found in RM chickens, are generally associated with increased enzymatic activity and enhanced nutrient absorption, owing to the larger surface area available for digestion [26]. For crypt depth, deeper crypts were observed in Ross chickens and are typically associated with increased cellular turnover and proliferation [27] which may reflect the fast growth rate of Ross chickens. Regarding the villus-to-crypt ratio, the highest value was in RM chickens and may indicate a more favourable balance between absorptive surface area and cellular turnover in their gut [28]. The lowest ratio in Ross, BP, and BP × SA chickens could indicate lower absorption capacity in favour of increased mucin secretion [28]. Previous reports show that within the same genotype, the height of the villi and the ratio of villi/crypts increase with age and live weight of chickens [29,30]. On the other hand, in our study, the observed changes in gut traits cannot be fully attributed to animal age (which differed between Ross and local breeds/crosses but was consistent within the latter) or live weight (which varied across the five genotypes) [22]. Although these factors may contribute to the differences among genotypes, further research is necessary to clarify their specific roles.

Additionally, an increased number of goblet cells, the highest in Ross chickens, which secrete protective mucus, can be considered a defence mechanism against environmental and dietary challenges [27,31]. Mucus production traps and neutralizes bacteria while providing niches for beneficial microbiota, contributing to gut health under stressful conditions [32,33]. This defence mechanism may be more pronounced in Ross chickens due to their sensitivity to environmental and dietary stressors. Furthermore, the growth curve results presented in previous research on the same animals [22] show the best results for local breeds and their hybrids and align with the findings of this study regarding gut morphology and intestinal immune response analysis.

Dietary input significantly impacted intestinal morphology. The deeper crypts found in chickens fed the low-input diet suggest increased tissue turnover, which could reflect an adaptive response to inflammation or dietary antinutritional factors (ANFs) [24,34]. Interestingly, the trend towards increased density of CD3+ cells in chickens fed the low-input diet may reflect a heightened local immune response, triggered by ANFs likely present in the low-input diet, as previously reported in pigs by Hampson, 1986 [24].

Based on preliminary results of the life cycle assessment (LCA), Ross chickens have a lower environmental impact than RM and BP. However, when chickens were fed a low-input diet, differences in environmental impact between Ross chickens and local chickens were reduced [35]. Given the challenges posed by environmental changes, identifying solutions that balance different aspects of animal husbandry is essential, with particular attention to animal welfare and the adaptability of different genotypes to challenging conditions.

Overall, the results suggest that RM chickens demonstrate greater resilience to dietary changes, exhibiting higher villus-to-crypt ratios and overall gut health. In contrast, fast-growing genotypes like Ross are optimized for high performance but may be less adaptable to low-input diets, as reflected in their intestinal morphology and immune responses. Although BP chickens are a slow-growing genotype and exhibit comparable crypt depth and goblet cell density to RM, for other parameters, such as villus height and villus-to-crypt ratio, they show a gut profile similar to Ross. As a result, BP chickens do not exhibit the same level of resilience to dietary changes as RM chickens. These findings highlight the complexity of the genotype–diet interaction and underscore the need for a nuanced approach to developing sustainable poultry farming systems that prioritize both productivity and welfare.

## 5. Conclusions

The current study analyzed the differences in gut morphology and inflammatory patterns of different genotypes and the effects of low-input diet on these parameters. While RM chickens demonstrated greater resilience to dietary changes compared to the fast-growing commercial genotype, not all local chicken breeds exhibited the same level of resilience. These findings highlight the potential of specific local breeds, such as RM, for sustainable poultry farming, particularly in systems designed to minimize environmental impact through the use of low-input diets. The results also emphasize the need for further research to explore the specific dietary components that affect gut health. Understanding the interactions between genotype, diet, and intestinal health would permit the refinement of poultry farming practises to balance productivity, environmental sustainability, and animal welfare.

## Figures and Tables

**Figure 1 animals-15-00696-f001:**
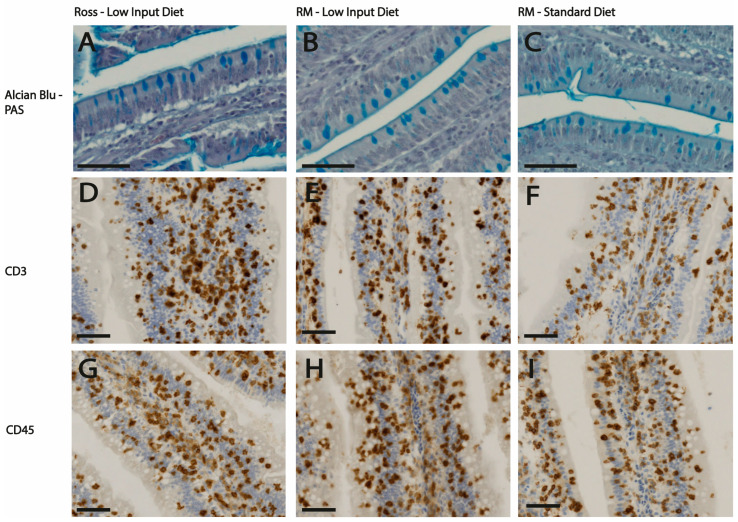
Representative light micrographs of jejunum sections from chickens fed either a low-input diet (**A**,**B**,**D**,**E**,**G**,**H**) or a standard diet (**C**,**F**,**I**), from two different genotypes: Ross (**A**,**D**,**G**) and RM (**B**,**C**,**E**,**F**,**H**,**I**). (**A**–**C**) show Alcian Blue-PAS-stained jejunal sections from Ross chickens (**A**) at 47 days of age and RM chickens (**B**,**C**) at 105 days of age. Goblet cells appear as Alcian Blue-positive. (**D**–**F**) show immunohistochemical staining for CD3+ cells, while (**G**–**I**) display CD45+ cells, both in jejunal sections from chickens fed the respective diets. Scale bars: 50 μm.

**Table 1 animals-15-00696-t001:** Effect of genotype, diets and interaction on jejunum morphology and gut inflammatory pattern in broiler.

	Chickens, No.	Villi Length, μm	Crypts Depth, μm	Villus/Crypt Ratio	Goblet Cells, No.	CD3+, Cells/10,000 μm^2^	CD45+, Cells/10,000 μm^2^
BP	12	963 ^A^	88.2 ^A^	11.1 ^AB^	19.2 ^A^	2943	3121
BP × SA	12	1016 ^A^	98.8 ^A^	10.5 ^AB^	19.4 ^A^	3678	3302
RM	12	1316 ^B^	90.5 ^A^	14.6 ^C^	17.7 ^A^	3257	3023
RM × SA	12	1167 ^AB^	106.4 ^A^	11.2 ^B^	17.9 ^A^	3296	3192
Ross	12	1028 ^A^	136.4 ^B^	7.8 ^A^	21.6 ^B^	3176	3291
Standard	30	1147	102.6	11.7 ^A^	18.9	3092	3198
Low input	30	1049	105.6	10.4 ^B^	19.5	3447	3174
*p* value (G)		<0.001	<0.001	<0.001	<0.001	0.280	0.806
*p* value (D)		0.046	0.529	0.039	0.172	0.095	0.888
*p* value (G × D)		0.189	0.190	0.163	0.280	0.369	0.368
RMSE		187	18.6	2.33	1.51	807	635

Means with different superscript letter are statistically different. Values are expressed as least square means with root mean square error (RMSE).

## Data Availability

The data presented in this study are available upon request from the corresponding author.

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
