# Peer review of "Histochemical and Immunohistochemical Evaluation of the Effects of a Low-Input Diet on Different Chicken Breeds"

_animals, 2025, doi:10.3390/ani15050696_

Round 1
Reviewer 1 Report
Comments and Suggestions for Authors
How could the lower growth capacity of local poultry breeds be balanced with environmental aspects?
Author Response
Thank you for the time you dedicated to reviewing our manuscript. We have revised the manuscript based on your and the other reviewers' suggestions. The changes are highlighted in red to make them easier to track. Additionally, we have included references (17, 18, 19, 20, 21) to better characterize the different genotypes, as requested.
Comment 1: How could the lower growth capacity of local poultry breeds be balanced with environmental aspects?
Response 1: Thank you for your revision. Generally talking, while the environmental impact of poultry meat produced in conventional intensive systems is recognized to be low compared to other animal products, the ecosystem services associated with the farming of slow-growing and local breeds can provide new benefits to specific areas that are not suitable for intensive and conventional production from a social, economic and environmental point of view.

Reviewer 2 Report
Comments and Suggestions for Authors
The article shows several interesting points such as the resilience of some broiler breeds (fast-growing breeds vs. slow-growing breeds, commercial hybrids vs. local breads), and their crosses with another slow-growing breed (Sasso) to dietary changes and their degree of adaptability. This adaptability is strategic for current production systems. The data obtained, in fact, highlight the complex genotype-diet interaction and how these characteristics must be taken into account to better approach sustainable breeding but with a careful "look" also at animal welfare.
That said, there are some points that the authors should modify and/or integrate.
Abstract. Lines 28-31. The authors state "This study evaluated the effects of diet (standard vs. low intake, formulated with reduced soybean meal in favor of local ingredients) on intestinal health in fast-growing chickens (Ross 308), local breeds (Bionda Piemontese. BP; Robusta Maculata, RM) and their crosses with Sasso hens (SA) (BP×SA, RM×SA)."
The manuscript focused on morphological/morphometric assessments and not on the effects of diet on intestinal health: this is inferable/debatable in the discussion section.
I suggest at the end of this sentence "on the morphological characteristics of the chicken/broiler chicken jejunum".
The authors used the Alcian blue + PAS method to stain goblet cells. Alcian blue, as well as PAS, stains a subpopulation of goblet cells (mucous cells that produce acidic or neutral mucin, respectively). In addition, mucus cells that express both mucins (acidic and neutral) show a purple staining (a combination of Alcian blue and PAS).
Still on the subject, I don’t understand why the authors didn’t report separate data for Alcian blue-positive, PAS-positive goblet cells (and cells that express both mucins). Moreover, why use a combination of Alcian blue + PAS and then report in Table 1 only a result that refers to a single count of a single subpopulation of goblet cells (only Alcian blue-positive cells? Only PAS-positive cells?)
The authors should explain why they use the CD45+ and CD3+ markers and which inflammatory cells they intend to stain (e.g. T lymphocytes, NK etc.).
The intraepithelial inflammatory cells were counted in the villi or in the crypts?
The method used to counted/evaluated the leukocytes is previously described or is original idea from authors?
In the "Statistical analysis" section, indicate how the values ​​are expressed (e.g. "the values ​​are expressed as mean ± standard deviation" or "the value are expressed as mean ± root mean square error RMSE"). Alternatively, below Table 1.
Pag.3 Line 101. Should be explain the acronym GMO-soybean (and in the other parts of the manuscript).
Pag.3 Lines 107-108. I suggest a x centimeters (e.g. about 10 cm?? About 20 cm?), near the Merckel diverticulum instead to "halfway between the end of the duodenal loop and the Merckel diverticulum".
Lines 196-198. “Furthermore, the growth curve results presented in previous research on the same animals [17] are in line with the results of this study regarding intestinal morphology and analysis of the intestinal immune response.” The authors should better explain this sentence because it is not clear how the data of the previous work are in line with this manuscript.
Lines 202-204. “Interestingly, the trend toward higher CD3+ cell density in chickens fed a low-intake diet may reflect an enhanced local immune response triggered by ANFs likely present in the low-intake diet [19].”. The authors should indicate that the cited article [19] (Hampson D.J. 1986) refers to studies in pigs: for example "as previously reported in pigs by Hampson, 1986".
Table1. I suggest separate the results in a different tables: in my opinion the morphometric evaluations (linear unit of measure, e.g. villi lenght and crypt depth) together the goblet cells number (number of mucous cells/ 300 μm) and inflammatory cells (surface measurement unit, inflammatory cells/10,000 μm2), should be separated each other. Alternatively you can divide the different assessments with a double line if the atuhors guideline allows it.
In Table 1, standardize the data obtained from the morphometric evaluations with one or two decimal places (where necessary).
Table 1. Check and standardize headers e.g. Villus/crypt ratio (final pt letters are on the next line).
Figure 1. The authors indicated in the Mat &Meth sections (as well as in the figure caption) that they used the combination of the Alcian blue + PAS method. The images are representative of the Alcian blue method only, they don’t show even one PAS positive goblet cell. How is this possible?
Figure 1. Standardize all images at the same magnification and check the scale bars (some are different both in longest and thickness).
Figure 1. The images C-D are unfocused. In addition, Image F is narrower than the others.
Author Response
We sincerely appreciate the time and effort you dedicated to reviewing our manuscript. In response to your and the other reviewers' suggestions, we have made the necessary revisions, with all changes highlighted in red for easier tracking. Additionally, we have added references (17, 18, 19, 20, 21) to further characterize the different genotypes, as requested.
Comment 1: Abstract. Lines 28-31. The authors state "This study evaluated the effects of diet (standard vs. low intake, formulated with reduced soybean meal in favor of local ingredients) on intestinal health in fast-growing chickens (Ross 308), local breeds (Bionda Piemontese. BP; Robusta Maculata, RM) and their crosses with Sasso hens (SA) (BP×SA, RM×SA)." The manuscript focused on morphological/morphometric assessments and not on the effects of diet on intestinal health: this is inferable/debatable in the discussion section. I suggest at the end of this sentence "on the morphological characteristics of the chicken/broiler chicken jejunum".
Response 1: Thank you for the suggestion. We agree with it and we changed as suggested. [Updated the manuscript, lines 29-30: This study evaluated the effects of the diet (standard vs. low-input, formulated with reduced soybean meal in favour of local ingredients) on the morphological characteristics of the jejunum in]
Comment 2: The authors used the Alcian blue + PAS method to stain goblet cells. Alcian blue, as well as PAS, stains a subpopulation of goblet cells (mucous cells that produce acidic or neutral mucin, respectively). In addition, mucus cells that express both mucins (acidic and neutral) show a purple staining (a combination of Alcian blue and PAS). Still on the subject, I don’t understand why the authors didn’t report separate data for Alcian blue-positive, PAS-positive goblet cells (and cells that express both mucins). Moreover, why use a combination of Alcian blue + PAS and then report in Table 1 only a result that refers to a single count of a single subpopulation of goblet cells (only Alcian blue-positive cells? Only PAS-positive cells?)
Response 2: Thank you for pointing this out, the positivity for PAS-positive cells were just a few in the whole count of the samples, a number that was not considerable alone, so the statistical analysis was performed only in the Alcian blu-positive cells. We corrected the description as reported: Figure 1. Representative light micrographs of jejunum sections from chickens fed either a low-input diet (panels A, B, D, E, G, H) or a standard diet (panels C, F, I), from two different genotypes: Ross (panels A, D, G) and RM (panels B, C, E, F, H, I). Panels A, B, and C show Alcian Blue-PAS-stained jejunal sections from Ross chickens (A) at 47 days of age and RM chickens (B, C) at 105 days of age. The goblet cells are Alcian blu-positive. [lines 187-193]
Comment 3: The authors should explain why they use the CD45+ and CD3+ markers and which inflammatory cells they intend to stain (e.g. T lymphocytes, NK etc.).
Response 3: As you suggested, we implemented the phrase as follow: were applied to detect CD45+ intraepithelial leukocytes, including T-cells and B-cells, and CD3+ intraepithelial T-cells in the jejunal mucosa [lines 150-151]
Comment 4: The intraepithelial inflammatory cells were counted in the villi or in the crypts?
Response 4: Thank you for the question, we counted the inflammatory cells in the villi, we corrected as follow: Using a reference rectangle with the short side at 100 μm, intraepithelial leukocytes in the villi were counted and represented as the density of CD45+ and CD3+ cells [Lines 154-155]
Comment 5: The method used to counted/evaluated the leukocytes is previously described or is original idea from authors?
Response 5: Thank you for asking, the method is an original idea from authors, as described in the following papers:
Huerta, A., Trocino, A., Birolo, M., Pascual, A., Bordignon, F., Radaelli, G., … Xiccato, G. (2022). Growth performance and gut response of broiler chickens fed diets supplemented with grape (Vitis vinifera L.) seed extract. Italian Journal of Animal Science, 21(1), 990–999. https://doi.org/10.1080/1828051X.2022.2084462
Zardinoni, G., Huerta, A., Boskovic Cabrol, M., Trocino, A., Fonsatti, E., Ballarin, C., … Xiccato, G. (2024). Gut response to the dietary supplementation with sodium butyrate in broiler chickens: morphology and microbiota composition and predictive functional groups. Italian Journal of Animal Science, 24(1), 123–136. https://doi.org/10.1080/1828051X.2024.2443481
Pascual, A., Pauletto, M., Trocino, A. et al. Effect of the dietary supplementation with extracts of chestnut wood and grape pomace on performance and jejunum response in female and male broiler chickens at different ages. J Animal Sci Biotechnol 13, 102 (2022). https://doi.org/10.1186/s40104-022-00736-w
Comment 6: In the "Statistical analysis" section, indicate how the values ​​are expressed (e.g. "the values ​​are expressed as mean ± standard deviation" or "the value are expressed as mean ± root mean square error RMSE"). Alternatively, below Table 1.
Response 6: As requested by the reviewer, the following text was added below table 1" Values are expressed as least square measns with root mean square error (RMSE)". [Lines 196-197]
Comment 7: Pag.3 Line 101. Should be explain the acronym GMO-soybean (and in the other parts of the manuscript).
Response 7: We have explained the acronym as follow: In the low input diet, imported genetically modified soybean (GMO-soybean) meal [Line 101]
Comment 8: Pag.3 Lines 107-108. I suggest a x centimeters (e.g. about 10 cm?? About 20 cm?), near the Merckel diverticulum instead to "halfway between the end of the duodenal loop and the Merckel diverticulum".
Response 8: Thank you for your suggestion. We chose to indicate anatomical landmarks instead of a specific length in centimeters due to potential individual and genotype-related variations. A morphological reference provides a more consistent and comparative measurement across different genotypes, ensuring accuracy in sample collection.
Comment 9: Lines 196-198. “Furthermore, the growth curve results presented in previous research on the same animals [17] are in line with the results of this study regarding intestinal morphology and analysis of the intestinal immune response.” The authors should better explain this sentence because it is not clear how the data of the previous work are in line with this manuscript.
Response 9: Thank you for the suggestion, we reported in the manuscript: Furthermore, the growth curve results presented in previous research on the same animals [22] where the best results were obtained in local breeds and their hybrids, align with the findings of this study regarding gut morphology and intestinal immune response analysis. [Line 221-222]
Comment 10: Lines 202-204. “Interestingly, the trend toward higher CD3+ cell density in chickens fed a low-intake diet may reflect an enhanced local immune response triggered by ANFs likely present in the low-intake diet [19].”. The authors should indicate that the cited article [19] (Hampson D.J. 1986) refers to studies in pigs: for example "as previously reported in pigs by Hampson, 1986".
Response 10: Thank you, we corrected as you proposed. Interestingly, the trend towards increased density of CD3+ cells in chickens fed the low input diet may reflect a heightened local immune response, triggered by ANFs likely present in the low input diet, as previously reported in pigs by Hampson, 1986 [24]. [lines 229-230]
Comment 11: Table1. I suggest separate the results in a different tables: in my opinion the morphometric evaluations (linear unit of measure, e.g. villi lenght and crypt depth) together the goblet cells number (number of mucous cells/ 300 μm) and inflammatory cells (surface measurement unit, inflammatory cells/10,000 μm2), should be separated each other. Alternatively you can divide the different assessments with a double line if the atuhors guideline allows it.
Response 11: Thank you, we added the double lines [lines 194-197].
Comment 12: In Table 1, standardize the data obtained from the morphometric evaluations with one or two decimal places (where necessary).
Response 12: We corrected with one decimal [lines 194-197].
Comment 13: Table 1. Check and standardize headers e.g. Villus/crypt ratio (final pt letters are on the next line).
Response 13: We corrected it [lines 194-197].
Comment 14: Figure 1. The authors indicated in the Mat &Meth sections (as well as in the figure caption) that they used the combination of the Alcian blue + PAS method. The images are representative of the Alcian blue method only, they don’t show even one PAS positive goblet cell. How is this possible?
Response 14: Thank you for asking, the PAS positive cells were rare in the samples, and not representative, almost all the goblet cells were Alcian Blu positive.
Comment 15: Figure 1. Standardize all images at the same magnification and check the scale bars (some are different both in longest and thickness).
Response 15: Thank you for the attention, we improved the standardization and the scale bars, only differences now are in the goblet cells and immunohistochemistry, to better identify the different measurements, also as requested by one reviewer we added a same breed-different diet column in the table to highlight the differences between the two diets.
Comment 16: Figure 1. The images C-D are unfocused. In addition, Image F is narrower than the others.
Response 16: Thank you, we improved the focus in the images.

Reviewer 3 Report
Comments and Suggestions for Authors
Reviewer comments
Title: Histochemical And Immunohistochemical Evaluation Of The 2 Effects Of A Low Input Diet On Different Chicken Breeds
Manuscript needs a minor revision for the following comments and suggestions:
Simple summary & Abstract
- Information on low-input diet is already mentioned in both sections. However, it is better to be consistent in these sections. Example: in lines17-18, you can say that GMO-soybean was replaced by local ingredients (including faba & GMO-free soybean meal), rather than mentioning only faba beans.
Introduction
- Brief information on the particular breeds, especially the local ones, can be included.
Materials and Method
This section needs some improvement as follows:
- Details on how many replicates/pen and how many birds per pen are important than mentioning the total number of both birds and pens.
- Why BP (37 vs 39), RM (25 vs 47) and BP×Sasso (49 vs 48) had different number of birds in each gender? Have you used all birds for this experiment or just 6 birds? Revise if needed.
- Mention about the common diet that has been fed until day 20.
- It is mentioned that half of the pens were fed standard diet, and the other half were fed low-input diet. How did you achieve this…were the birds reallocated into separate pens or by other means? Please check and revise as needed.
- How did you decide on different slaughtering-days for Ross (47 d) and others (105 d)? Need some explanation.
Results
- Line 156: Villus height 1179 µm mentioned here is different from that of Table 1 (1147 µm). Check and revise.
- Lines 161-162: There is no Table 2. Revise it to Table 1.
Table
- Superscripts (A/B) are missing for the main effects (P=0.039) of diet for Villus/crypt ratio (Table 1).
Figures
- Why did you take 2 different breeds (Ross & RM) to illustrate the difference between 2 diets? Is it possible to include figures on “same breed & different diets” or “different breeds & same diet”?
Author Response
Thank you for your valuable feedback and the time you devoted to reviewing our manuscript. We have carefully implemented the suggested revisions, and all modifications are highlighted in red for clarity. Furthermore, we have incorporated additional references (17, 18, 19, 20, 21) to enhance the characterization of the different genotypes.
Manuscript needs a minor revision for the following comments and suggestions:
Simple summary & Abstract
Comment 1: Information on low-input diet is already mentioned in both sections. However, it is better to be consistent in these sections. Example: in lines17-18, you can say that GMO-soybean was replaced by local ingredients (including faba & GMO-free soybean meal), rather than mentioning only faba beans.
Response 1: Thank you, we corrected following your suggestion: a standard diet and a low-input diet that replaced part of the soybean content with local ingredients (including fava beans & GMO-free soybean). [Line 18]
Introduction
Comment 2: Brief information on the particular breeds, especially the local ones, can be included.
Response 2: Thank you for your suggestion, we integrated the manuscript as follow: The local breeds considered in this study were Bionda Piemontese (BP) and Robusta Maculata (RM). Bionda Piemontese, originating from the Piemonte region in Italy, is a slow-growing, late-maturing breed [17]. Although primarily farmed for meat production, it is considered a dual-purpose breed [18]. Robusta Maculata was developed from crosses between Tawny Orpingtons and White Americans and is used for both meat and egg production [19]. Hendrix Genetics produced the meat-type chicken known as the Sasso in France and brought it to market. The sasso chickens are renowned for their medium growth rate, medium size, and soft flesh. Additionally, due to its recessive plumage traits, this chicken genotype is frequently utilized in crossbreeding programs with regional breeds [20]. Due to its exceptional development rate and feed efficiency, the Ross 308 is the most popular genotype of fast-growing chickens raised for meat production worldwide [21]. [Lines 76-87]
Materials and Method
This section needs some improvement as follows:
Comment 3: Details on how many replicates/pen and how many birds per pen are important than mentioning the total number of both birds and pens.
Response 3: Thank you for the suggestion, we added this detail as follow: i.e. two replicate pens per experimental group (genotype/sex/diet). [lines 110-111]
Comment 4: Why BP (37 vs 39), RM (25 vs 47) and BP×Sasso (49 vs 48) had different number of birds in each gender? Have you used all birds for this experiment or just 6 birds? Revise if needed.
Response 4: We revised as follow: Two days before commercial slaughtering, 6 males per genotype per experimental diet were used to sample jejunum mucosa and perform the analyses presented in this study. [lines 128-129]
Comment 5: Mention about the common diet that has been fed until day 20.
Response 5: Thank you for the suggestion, we corrected as follow: The animals were fed a starter diet until 20 days of age; the details of the starter diet are presented in Menchetti et al. [22]. [lines 11-112]
Comment 6: It is mentioned that half of the pens were fed standard diet, and the other half were fed low-input diet. How did you achieve this…were the birds reallocated into separate pens or by other means? Please check and revise as needed.
Response 6: Thank you. To avoid any misunderstanding the text was mofidied as follows: The trial was conducted in the experimental poultry house facilities of the Univer-sity of Padova (Italy) where 40 pens were used in which a total of 441 chickens were allocated according to a three-factorial experimental arrangement with five genotypes and two sexes (Ross, 51 females and 51 males; Bionda Piemontese BP, 37 and 39; Robusta Maculata RM, 25 and 47; BP×Sasso, 49 and 48; and RM×Sasso, 47 and 47) and two diets (Standard Diet: metabolizable energy, ME 3,348 kcal/kg; crude protein, CP 18.5%; Low-input Diet: ME 3,084 kcal/kg; CP 16.7%), i.e. two replicate pens per experimental group (genotype/sex/diet) [22] . The animals were fed a starter diet until 20 days of age; the details of the starter diet are presented in Menchetti et al. [22]. The experimental diets were fed from 20 days of age until slaughtering (47 days for Ross and 105 days for other genotypes). [Lines 104-113]
Comment 7: How did you decide on different slaughtering-days for Ross (47 d) and others (105 d)? Need some explanation.
Response 7: Thank you, we corrected as follow: The different slaughter ages for Ross (47 days) and the other genotypes (105 days) were determined based on their distinct growth rates and physiological development [22]. Given the experimental conditions in this study, extending the trial duration for Ross beyond its typical commercial slaughter age was too challenging. Therefore, Ross 308 was slaughtered at 47 days, aligning with standard industry practices and achieving approximately 55% of its adult body weight. Conversely, local breeds and their cross-breeds exhibit significantly slower growth rates, requiring more time to reach physiological maturity. These birds were slaughtered at 105 days, ensuring they reached a comparable maturity level of 55%–65% of their adult body weight [22]. [lines 117-125]
Results
Comment 8: Line 156: Villus height 1179 µm mentioned here is different from that of Table 1 (1147 µm). Check and revise.
Response 8: thank you for notice this, we checked it and corrected.
Comment 9: lines161-162: There is no Table 2. Revise it to Table 1.
Response 9: thank you, we modified it.
Table
Comment 10: Superscripts (A/B) are missing for the main effects (P=0.039) of diet for Villus/crypt ratio (Table 1).
Response 10: thank you, we added the two superscripts as suggested.
Figures
Comment 11: Why did you take 2 different breeds (Ross & RM) to illustrate the difference between 2 diets? Is it possible to include figures on “same breed & different diets” or “different breeds & same diet”?
Response 11: Thank you for asking, we chose to illustrate the difference between the two situations with most variety among the effects (genotypes and diet), so the two breeds that show more differences in the two different diets. To improve the images we follow your indications adding the figures also of “same breed & different diets”. [Fig. 1, Line 185]

Reviewer 4 Report
Comments and Suggestions for Authors
This manuscript describes a trial with reduced environmental impact, reduced energy and protein in chickens. However, authors provide details on current study only on histology/histomorphometry of the gut. The performance data, and diets composition and LCA methodology is very important for such an evaluation.
Author Response
Thank you for the time you dedicated to reviewing our manuscript. We have revised the manuscript based on your and the other reviewers' suggestions. The changes are highlighted in red to make them easier to track. Additionally, we have included references (17, 18, 19, 20, 21) to better characterize the different genotypes, as requested, and 35 for the LCA methodology.
Thank you for your time and valuable suggestions. The performance data and diet compositions are described in greater detail in the following study:
Menchetti, L.; Birolo, M.; Mugnai, C.; Cartoni Mancinelli, A.; Xiccato, G.; Trocino, A.; Castellini, C. Effect of genotype and nutritional and environmental challenges on growth curve dynamics of broiler chickens. Poult. Sci. 2024, 103, 104095. https://doi.org/10.1016/j.psj.2024.104095.
As suggested, we have implemented additional references in the manuscript, specifically in the following section:
"The trial was conducted in the experimental poultry house facilities of the University of Padova (Italy), where 40 pens were used to allocate a total of 441 chickens according to a three-factorial experimental design, considering five genotypes and two sexes (Ross: 51 females and 51 males; Bionda Piemontese (BP): 37 and 39; Robusta Maculata (RM): 25 and 47; BP×Sasso: 49 and 48; and RM×Sasso: 47 and 47), and two diets (Standard Diet: metabolizable energy, ME 3,348 kcal/kg; crude protein, CP 18.5%; Low-input Diet: ME 3,084 kcal/kg; CP 16.7%), with two replicate pens per experimental group (genotype/sex/diet) [22]. The animals were fed a starter diet until 20 days of age; details of the starter diet are presented in Menchetti et al. [22]. The experimental diets were provided from 20 days of age until slaughtering (47 days for Ross and 105 days for the other genotypes). In the low-input diet, imported genetically modified soybean (GMO-soybean) meal was partially replaced with local ingredients, such as faba bean (Vicia faba, var. minor) and GMO-free organic soybean meal. Further details on diet composition and other recorded parameters can be found in Menchetti et al. [17]." [Lines 104-117]
Regarding the LCA methodology, we have incorporated an additional reference in the discussion and modified the manuscript as follows: “Based on preliminary results of life cycle assessment (LCA), Ross chickens have a lower environmental impact than RM and BP. However, when chickens were fed a low-input diet, differences in environmental impact between Ross chickens and local chickens were reduced [35]. Given the challenges posed by environmental changes, identifying solutions that balance different aspects of animal husbandry is essential, with particular attention to animal welfare and the adaptability of different genotypes to challenging conditions.” [Lines 230-236]
[35] Berton, M.; Huerta, A.; Trocino, A.; Bordignon, F.; Sturaro, E.; Xiccato, G.; Birolo, M. Life Cycle Assessment of Broiler Chicken Production Using Different Genotypes and Low-Input Diets. In Book of Abstracts of the 73rd Annual Meeting of the European Federation of Animal Science, Proceedings of the 73rd EAAP Annual Meeting, Porto, Portugal, 5–9 September 2022; Wageningen Academic Publishers: Wageningen, The Netherlands, 2022; No. 28.

Round 2
Reviewer 4 Report
Comments and Suggestions for Authors
Authors revised their work adequately.